# Health-Related Quality of Life of Medical Students in a Chinese University: A Cross-Sectional Study

**DOI:** 10.3390/ijerph16245165

**Published:** 2019-12-17

**Authors:** Yanli Qiu, Mingkang Yao, Yiwei Guo, Xiaowei Zhang, Shuoyang Zhang, Yuting Zhang, Yixiang Huang, Lingling Zhang

**Affiliations:** 1Zhongshan School of Medicine, Sun Yat-sen University, Guangzhou 510080, China; qyanli@mail2.sysu.edu.cn (Y.Q.); mkyao528360@163.com (M.Y.); cipherblaze2@gmail.com (Y.G.); zhangxw1997@gmail.com (X.Z.); zsy15521170771@163.com (S.Z.); zhangyt43@mail2.sysu.edu.cn (Y.Z.); 2School of Public Health, Sun Yat-sen University, Guangzhou 510080, China; 3Department of Nursing, College of Nursing and Health Sciences, University of Massachusetts, Boston, MA 02125-3393, USA; lingling.zhang@umb.edu

**Keywords:** health-related quality of life, SF-36, medical students, China

## Abstract

Thus far, there have been no studies adapting the Mandarin 36-Item Short Form Health Survey (the SF-36) questionnaire for assessment of the health-related quality of life (HRQOL) of medical students in China. This study aimed to explore the feasibility of that form and analyse its impact factors. The study involved 498 randomly sampled medical students stratified by their academic majors, and general information was collected. The effective response rate was 83.53%. Split-half reliability coefficients and Cronbach’s α coefficients of seven dimensions were more than 0.7 with the exception of the social function (SF) dimension. Spearman’s correlation analysis results were basically in accord with the theoretical construction of the SF-36. The HRQOL of the students was scored from 43.83 (the RE dimension) to 93.34 (the PF dimension). The primary impact factors affecting the HRQOL of medical students included major, sleep quality, degree of physical exercise, post-exercise status, relationship with roommate, and satisfaction with family. These findings suggested that the Mandarin SF-36 was reliable for measuring the HRQOL, that the HRQOL of medical students in a Chinese university was relatively poor, and that its improvement requires concerted efforts.

## 1. Introduction

The process of medical education has long been proved to be a challenging scenario that threatens medical students’ physical and psychological health [1]. Compared with students of the same age in other majors, medical students must bear higher occupational stress in adapting to their new lifestyle and are often exposed to patients’ deaths [2]. The great academic pressure, severe daytime somnolence, lack of physical exercise and social interaction, as well as heavy stress on scientific research [3] combined can lead to a decreasing quality of life for medical students. A systemic survey suggests that medical students show higher levels of psychological anxiety, depression, and distress [4].

The World Health Organization has defined the quality of life as the perception of people’s position in life in the context of the culture and value systems in which they live and in relation to their goals, expectations, standards, and concerns [5]. It is a broad, multidimensional and polysemic concept affected in a complex way by the person’s physical health, psychological state, degree of independence, social relationships and relationship to salient features of the individual’s environment [6]. It is important for medical schools and educators to understand the health-related quality of life (HRQOL) of students during their medical training [7], and research on HRQOL potentially facilitates the transformation of curriculum design and may lead to the promotion of physical and mental health of students.

The present study is innovative in that it first adopted the measurement of the 36-Item Short Form Health Survey (the SF-36) questionnaire to evaluate the QOL of Chinese medical students. In 1991, the international quality-of-life assessment formulated the standard procedures of the SF-36 to unify its use in various countries. The SF-36 questionnaire was derived from the Medical Outcomes Study (MOS) in the Boston Health Research Institute of the United States in 1989, an instrument with 149 items [8]. The initial objective of the MOS was to evaluate the medical decisions and patient outcomes under the influence of different systematic health care approaches. However, its simplified version was extended to assess the health-related quality of life and finally became the SF-36 questionnaire used today. Since then, the SF-36 questionnaire has been a widely validated and popular tool used in the assessment of quality of life among the general population ages 14 years and older [9]. WHOQOL-BREF and SF-36 are the most adopted instruments in assessing QOL of the general population, although Krageloh et al. showed that some items on the WHOQOL-BREF may not be suitable for medical students [10]. Further, Huang et al. showed that the SF-36 had better discrimination among different levels of health status than the WHOQOL-BREF in measurement of the impact of interventions on health-related QOL [11]. The Mandarin SF-36 has been widely used in the QOL measurement of general Chinese populations [12,13,14], and the reliability and validity of this form in Chinese medical students have been verified [9]. Therefore, the need to assess QOL of medical students by means of the SF-36 is justified.

The primary objective of this study was to explore the feasibility of the Mandarin version of the SF-36 in assessing the health-related quality of life (HRQOL) of medical students and to assess the HRQOL of medical students at Sun Yat-sen University, while aiming to find the associated factors that caused the variations among these students.

## 2. Materials and Methods

### 2.1. Sample

From 28 June 2018 to 22 July 2018, a cross-sectional study was conducted among medical students with majors in clinical medicine, oral medicine, and preventive medicine at Sun Yat-sen University, Guangzhou, China. A total sample size of 498 students was conveniently defined. It should be noted that the Chinese educational system and the courses of study vary depending on the majors and branches. In a typical medical school, clinical medicine usually has two branches: a five-year medical programme and an eight-year medical programme. The curriculum of the five-year programme includes one year of fundamental science, two years of basic medicine, one year of clinical medicine, and one year of internship, and provides a bachelor’s degree to its participants. Compared with the five-year programme, the eight-year medical programme adds an additional year of fundamental science and two years of rotation and offers a MD degree. The courses received by five-year medical programme students in the 3rd year mirror those of eight-year medical programme students in the 4th year. This provided us with an approach for exploring potential outcomes of different students learning the same subjects. Subsequently, we also considered that we should compare not just medical students of similar curricula, but also those of different curricula. As a result, a cluster sampling procedure was applied according to five different majors and years: eight-year medical programme students in the 4th year, preventive medicine in the 4th year, oral medicine in the 4th year, and five-year medical programme students in the 3rd and 4th years. We randomly selected three classes from each group mentioned above to form the sample. Students who dropped out or changed major were excluded, since they would not be qualified as sufficiently effective samples.

### 2.2. Measures and Procedures

To establish our sample pool, we invited students to complete a questionnaire consisting of the Chinese version of the SF-36 and general items. The Chinese version of the SF-36 measures eight health-related domains, including physical function (PF, limitations in physical activities because of health problems), physical role (RP, limitations in usual role activities because of physical health problems), body pain (BP), general health (GH, general health perceptions), vitality (VT, energy and fatigue), social function (SF, limitations in social activities because of physical or emotional problems), emotional role (RE, limitations in usual role activities because of emotional problems), mental health (MH, psychological distress and well-being) and one single-item dimension on health transition [8]. The physical component summary (PCS) consists of PF, RP, BP and GH, while the mental component summary (MCS) consists of VT, SF, RE and MH. The general items mainly included gender, major, exercise level, family status, sleep quality and social contact, which have possible links to the QOL. Data collection was carried out during class intervals or after classes in online and offline forms. The different forms of the questionnaire, either online or paper-based, had little impact on the equivalence of our study [15]. Students’ participation was voluntary, and the survey was anonymised. When a student returned a questionnaire with incomplete information, he/she was immediately invited back to complete the survey. If more than half of the domains were answered, the questionnaire was deemed effective unless the general items were not answered. The missing values of an effective questionnaire would then be replaced by the mean score of the item. Questions in the SF-36 included those about the health-related quality of life matters that had occurred in the preceding four weeks. All participants gave their informed consent for inclusion before being accepted into the study. The study was conducted in accordance with the Declaration of Helsinki, and the protocol was approved by the Ethics Committee of the school of public health at Sun Yat-sen University.

### 2.3. Data Analysis

The survey was administered as samples, and measures were well-prepared. The flow chart for data collection is shown in Figure 1. Data collected from the SF-36 were formulated in eight domains: physical function, role limitations due to physical problems, bodily pain, general health perceptions, vitality, social function, role limitations due to emotional problems and mental health. The raw scores of the eight domains were processed and transformed to a value from 0 to 100 according to the following Excel (Microsoft, Seattle, USA) software formula:(1)$$\mathrm{Transformed}\;\mathrm{score}=\frac{Actual\;raw\;score-Lowest\;possible\;raw\;score}{Largest\;possible\;raw\;score\;range}\times 100$$

The PCS was the mean transformed score of physical function (PF), physical role (RP), body pain (BP) and general health (GH), and the MCS was the mean transformed score of vitality (VT), SF, emotional role (RE) and mental health (MH). Higher scores suggested a higher level for the QoL. The reliability and correlation of the SF-36 were evaluated through split-half reliability, Cronbach’s α coefficients and Spearman’s correlation coefficients. Split-half reliability was calculated by comparing the scores of the odd half with those of the even half in each SF-36 dimension. Cronbach’s α coefficient assessed the internal consistency of the SF-36 questionnaire and a Cronbach’s α coefficient not less than 0.7 was generally considered sufficient to demonstrate internal consistency [16]. In addition, Spearman’s correlation coefficients reflected the correlation of eight dimensions and two aspects of the HRQOL.

An *f* test was also calculated to evaluate the homogeneity of variance. Based on the homogeneity of variance, one-way analysis of variance and a *t* test were applied to compare the means of the SF-36 and its component scores according to different impact factors. Otherwise, the Kruskal-Wallis test would be applied. We set α = 0.1 to avoid missing potential important factors. Impact factors differing among at least one of the eight dimensions were selected to make an analysis of multivariate stepwise regression in the corresponding dimensions. Data were analysed by a set of programmes provided by SPSS Statistics, version 21.0.24 (IBM, Armonk, USA). The data analysis adopted a form of the double-analysis method, meaning that two people analysed the same data, to avoid errors caused by anthropic factors in the analytical process.

## 3. Results

Of the 498 randomly selected students, 422 completed and returned questionnaires with the requested SF-36 data. Among the 422 questionnaires, six were removed for incomplete data on general items. The effective responses of the questionnaire including SF-36 and general items were 416, i.e., 83.53% of all participants.

Seven of the eight SF-36 dimensions (PF, RP, BP, GH, VT, RE and MH) had the split-half reliability coefficient higher than 0.8, except the SF dimension with a value of 0.516. The internal consistency of the SF-36 items was assessed by Cronbach’s α coefficients, ranging from 0.481 (the SF dimension) to 0.879 (the BP dimension). Similar to the spit-half reliability coefficients, Cronbach’s α coefficients of seven dimensions including PF, RP, BP, GH, VT, RE and MH were not less than 0.7 except for the SF dimension. Spearman’s correlation analysis further showed that PF, RP, BP and GH were correlated with PCS, while VT, SF, RE and MH were correlated mainly with MCS, results that were basically in accord with the theoretical construction of SF-36. Among the eight dimensions, RP and RE were the best measures of the PCS and MCS, respectively (Table 1).

Table 2 shows the social demographic characteristics of medical students in our study by gender and major. The means of eight dimensions between males and females did not show a significant difference (*p* > 0.1). The means of different majors showed a significant difference among Physical Role (RP), Body Pain (BP), General Health (GH), Vitality (VT), Social Function (SF) and Mental Health (MH) (*p* < 0.1). The means of eight dimensions among medical students in the study ranged from 43.83 (the RE dimension) to 93.34 (the PF dimension). The RE dimension, as the best measurement of MCS, scored the lowest among the eight dimensions of medical students in the study. As the best measure of PCS, the RP dimension scored 70.00.

Table 3 and Figure 2 compare the eight SF-36 dimension scores among different Chinese populations, including American Chinese [17], Taiwanese [18], Hong Kong [19], Sichuan [12], Hangzhou [14] and Shanghai [13]. Medical students involved in the survey scored highest in the PF dimension and lowest in the RE dimensions. 

The MH dimension score of medical students was close to that of the Hangzhou population, ranking the lowest. The SF, BP and RP dimension scores of Taiwanese, Hong Kong, Sichuan, Hangzhou and Shanghai populations were higher than those of medical students involved in the survey, who scored lower than Taiwanese, Sichuan, and Shanghai populations in the VT and GH dimensions.

Table 4 shows scores of medical students according to different impact factors of QOL in the study. The HRQOL scores of different physical exercise times per week showed a significant difference among the dimensions of PF, GH, VT and MH (*p* < 0.1). Post-exercise status, indicating participants’ subjective feelings of both physical and mental status after exercise, had a significant influence on the dimensions of PF, RP, BP, GH, VT and MH. Different levels of relationships with roommates scored significantly differently in all dimensions except PF. The level of satisfaction with family significantly influenced almost all dimensions except RP. The means of eight dimensions among different sleep qualities all showed a significant difference. Distressing family events in one year had a significant influence on RP. No significant differences were observed among different times for extracurricular research (*p* > 0.1).

Table 5 shows standardised regression coefficients of the impact factors on quality of life resulting from multivariate stepwise regression, which was calculated based on the outcomes in Table 3. It was observed that sleep quality was evidently the main and relatively strong positive impact factor influencing the SF-36 dimensions, whose standardised regression coefficient was the maximum for most SF-36 dimensions. Physical exercise level showed positive association with the GH dimension. Post-exercise status showed a positive influence on the PF, RP, GH, VT and MH dimensions. The RP, VT, SF and MH dimensions were positively associated with improving relationships with roommates. Increasing satisfaction with family had a positive influence on the BP, GH, VT, SF and MH dimensions. Medical students in preventive medicine in the 4th year showed a higher quality of life in the BP, VT and MH dimensions, and those in oral medicine in the 4th year showed a higher quality of life in the GH, VT and MH dimensions. It was evident that medical students in preventive medicine and oral medicine generally had a higher quality of life than students in clinical medicine (students of the eight-year programme in the 4th year, the five-year programme in the 3rd and 4th years). Further, the quality of life of students in clinical medicine (five-year programme) in both the 3rd and 4th years was not significantly different from that of students in clinical medicine (eight-year programme) in the 4th year.

## 4. Discussion

As one of the most popular instruments available for assessing the QOL of the general population, the SF-36 has been applied nearly universally. So it is no surprise that researchers have been using the Mandarin version of the SF-36 to evaluate segments of the general population and those with chronic diseases in China ever since it was first translated in 1991 [12,13,14,20,21]. The quality of life of medical students, who play an important role in the future of medicine, deserves better attention. Our study assessed the reliability and correlation of the SF-36 in measuring the quality of life of medical students. The results indicated that the health-related quality of life among medical students at Sun Yat-sen University was generally poor. According to the research data, this can be mainly attributed to deteriorations in sleep quality, tense relationships with roommates, unstable satisfaction levels with family, bad exercise status, and different majors.

In seven of eight dimensions, the spilt-half reliability coefficients valued more than 0.8 and Cronbach’s α coefficients valued more than 0.7. This showed a good internal consistency of the SF-36 for the purposes of the study. But the SF dimension had a low split-half reliability coefficient and Cronbach’s α coefficient of less than 0.7. Previous studies have reported similar issues [13]. It may be the result of an unclear conceptualization of social function in the Mandarin SF-36 and certain misunderstandings caused by differences in cultures. In China, “social activities”, refer not only to everyday life with someone with whom they are familiar in informal situations, but also to formal activities with other people, such as joining a new department or attending a conference. Individuals occasionally find themselves required to participate in formal activities in the face of a certain degree of ill health or a bad mood. Misunderstandings may result in low Cronbach’s α levels. Spearman’s correlation coefficient indicated a good correlation of PCS and MCS with their respective dimensions. Taking all those factors into consideration, we can conclude that the Mandarin version of the SF-36 has good reliability and correlation for measuring the quality of life of the medical students involved.

Our study found that the quality of life of medical students was relatively poor when compared with each dimension of the SF-36 in different Chinese populations. Except for the highest PF dimension score (93.34), the medical students had the lowest scores in the RE and MH dimensions, relatively lower scores in the SF/BP/RP dimensions, and some moderate scores in the VT/GH dimensions. Intriguingly, though the medical students had the highest score in the PF dimension. This may be influenced by the research population in the survey, consisting of college students, who tend to have higher physical function than other populations, including the elderly. The RE dimension, which is often considered one of the best measures of MCS, scored the lowest among the eight dimensions in our study, at 43.83 (43.04). The value was lower than that of other Chinese populations, and this fact demonstrated that the quality of life of medical students in our study was fairly poor. This may be attributed to high workloads, heavy academic pressure, a lower household income, and low sleep quality. Zhong et al. reported that one-third of medical students undergoing postgraduate neurology specialty training in China showed symptoms of depression, and those without such symptoms had significantly higher QoL scores [22]. Accurate measures should be taken by medical schools to ensure emotional support for their students, thus improving the QoL scores.

The factor of gender was proven to be irrelevant, as no significant differences in the quality of life were found between males and females. Paro and Domantay reported lower HRQOL scores for female medical students in Brazil and The Philippines, respectively [3,23]. This conflicting result is considered to have ties with cultures and different social climates. The fact that several foreign studies reported lower HRQOL in female medical students may be attributed to a culture that encourages stronger masculinity, in which men are fostered to appear less emotional and to hide their feelings and weaknesses [24,25]. There is, however, an alternate way of thinking, which was seen in a previous study about empathy in genders [26], whereby the one-child policy and the highly competitive process that begins in elementary school and lasts all the way to leading medical schools in China could foster more assertive, more independent, and psychologically stronger girls. Such thinking might help explain the contradictions between our study’s findings and those of studies outside China.

It was evident that students in the 4th year of clinical medicine in both the eight-year and five-year programmes had a lower quality of life in the BP, GH, VT and MH dimensions than did those in the same year of preventive medicine and oral medicine. It should be taken into account that different majors had different curricula and study workloads, which could contribute to the differences in HRQOL. One hint of this difference in QOL might be that, in our study, medical students in clinical medicine are studying either basic medicine or clinical medicine course s, like pathology, microbiology, and parasitology. All of these courses are known for placing heavy study burdens on their participants. Compared with the drastic situation of those students in clinical medicine, students in the 4th year of both preventive medicine and oral medicine have already finished their basic and clinical medicine studies. Students in preventive medicine begin internships and specialised courses, respectively, in the first and second semesters of the 4th year, while students in oral medicine study their specialised course in the entire 4th year. This means a reduced academic burden, which might explain the differences in QOL scores. A previous study reported that the presence of hypertension was correlated with reduced fitness [27]. The heavy study burden of clinical medicine students in learning basic and clinical medicine should be heeded, and effective intervention should be undertaken. Selvaraj and Bhat found that the development of positive psychological strengths such as hope, efficacy, resilience, and optimism within college students significantly improved their mental health condition [28]. Additionally, interventions, like avoiding excessive concentrations of complicated courses or adopting new teaching and examination methods, are another option. Flipped classrooms, an increasingly popular teaching method, have been demonstrated to improve medical students’ learning motivation and performance on examinations, while their effect on HRQOL was not clear [29,30]. But another study also showed that flipped classrooms might increase the burden and pressure for students, so their actual effects on medical education need further insight [31]. Thus, more studies are required to explore different teaching methods that can improve the quality of life of medical students on the basis of guaranteeing their professional abilities and basic devotion to the medical cause.

It is no surprise that, according to the results of our study, the degree of exercise had a positive impact on the GH dimension. The evidence of this can also be found in previous studies [32,33]. Among those studies are some systematic reviews and meta-analyses, in which enhanced HRQOL was demonstrated to have links with physical activities [34,35]. Therefore, it is unfortunate that, according to our study, 44.47% of medical students never exercise, and 36.54% of them exercise only 1–2 times per week. Thus, it is of great importance for medical schools to develop strategies that motivate regular physical activity in their students. This approach could improve the quality of life of medical students and may enable them to give patients better consultation on physical activities that can help in rehabilitation and disease prevention [36]. The study also found that quality of life was associated with post-exercise status, which can be defined as a feeling of tiredness and transient inability to maintain optimal physical performance [37]. The feeling of tiredness and cognitive concomitants such as “not wanting to continue or initiate a task” could appear after prolonged periods of cognitive activity after exercise [38]. Hence, it is suggested that students do physical activities within a suitable range and exercise regularly to improve their tolerance of such activities.

Sleep quality was found to be the main and more defining positive factor for the SF-36 dimensions in our study. This was demonstrated in several recent studies that underscored the strong relationship between higher sleep quality and higher QOL scores [39,40]. There is no doubt that sleep affects many aspects of bodily function [41]. From cognitive performance and mood to our immune system, the effects of sleep can be seen clearly [42,43]. To improve sleep quality, one should avoid poor sleep habits, such as Internet surfing at night, excessive daytime sleepiness and bad eating habits [44,45]. Food with a high glycemic index consumed more than 1 h before bedtime, small doses of tryptophan, and the hormone melatonin may contribute to the improvement of sleep quality [46]. It has also been found in some studies that physical exercise may also increase sleep quality [47].

The results of our study imply that better relationships with roommates are associated with an improved quality of life by affecting the RP, VT, SF and MH dimensions. To deal well with roommates, one should be more empathic and compromise appropriately, and regular organisation of dormitory group activities may be beneficial. In our questionnaire, the standard of satisfaction with family took economic conditions, cultural environment and relationships with family members into consideration. Satisfaction with family had links with a higher quality of life, suggesting that family environment may play an important role in the quality of life of medical students. While no significant difference in quality of life was observed between medical students who suffered distressing family events in one year and those who did not, this may relate to how family members cope with such things, but this remains unknown. Further studies about the influence of family environment on the quality of life of medical students should be undertaken.

### Strengths and Limitations of This Study

Certain flaws remained. For instance, detailed information on non-responders was not collected, nor were we sure whether there were differences between responders and non-responders. Further, our sample had an adequate size but was recruited from only one university due to limited funds, which also limited the generalisation of our results, and further studies need to be done. Moreover, due to the inherent limits of a cross-sectional study, timing between independent and dependent variables was not under strict control. It should also be noted that certain option-setting about some impact factors focused more on participants’ subjective feelings than on precise and objective indices, such as the level of exercise and sleep quality. This kind of option-setting benefited the participants to better complete the questionnaires in a limited time. Further studies on the quality of life are still required, where more precise and objective indices can be applied.

## 5. Conclusions

It has been demonstrated through our data and analysis that the Mandarin version of the SF-36 was a reliable tool for measuring HRQOL and that the health-related quality of life among medical students at Sun Yat-sen University was relatively poor. With respect to the data, the HRQOL relates mainly to major, sleep quality, degree of physical exercise, post-exercise status, relationships with roommates and satisfaction with family. The results of our study suggest that to increase the quality of life of medical students, certain interventions should be carried out by the education authorities to decrease academic pressure and divert more efforts to physical education for medical students. It should also be pointed out that sufficient personal care and emotional support should be provided by family members. Reflecting upon ourselves, we also believe that medical students themselves should pay more attention to their quality of life and take action to improve their own HRQOL.

## Figures and Tables

**Figure 1 ijerph-16-05165-f001:**
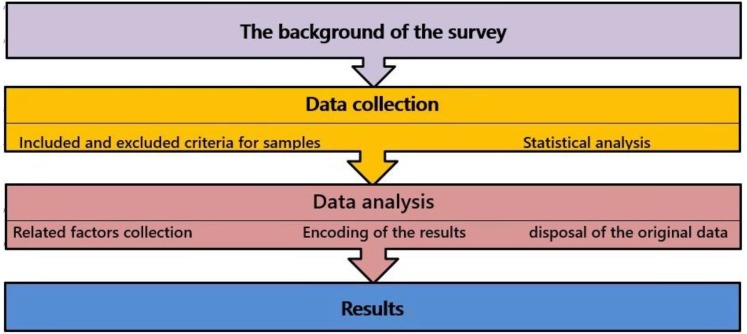
Flow chart of data collection, organisation, and analysis of data.

**Figure 2 ijerph-16-05165-f002:**
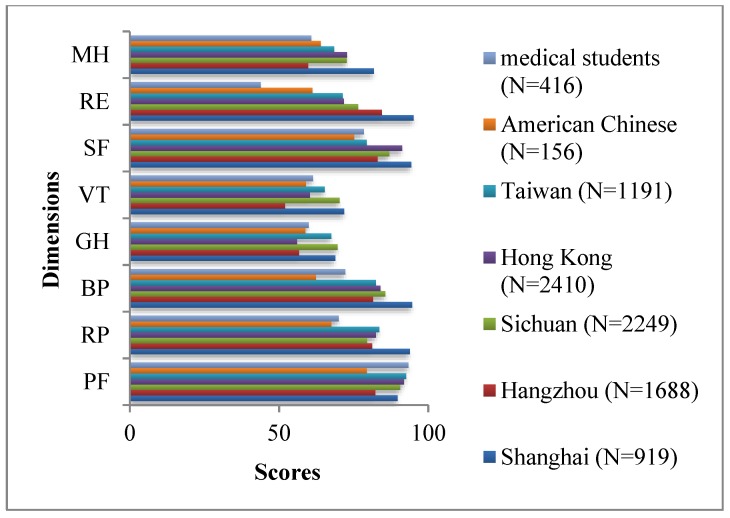
Scores of different Chinese populations.

**Table 1 ijerph-16-05165-t001:** Reliability and correlation of the SF-36 dimensions.

Dimension	Item Number	Reliability		Spearman’s Correlation Coefficient	
		Split-Half Reliability	Cronbach’s α Coefficient	PCS	MCS
PF	10	0.894	0.813	0.481	0.240
RP	4	0.877	0.864	0.863	0.444
BP	2	0.897	0.879	0.706	0.328
GH	5	0.851	0.839	0.740	0.472
VT	4	0.841	0.713	0.362	0.716
SF	2	0.516	0.481	0.452	0.691
RE	3	0.846	0.839	0.414	0.867
MH	5	0.829	0.770	0.393	0.754

**Table 2 ijerph-16-05165-t002:** Social demographic characteristics of medical students in the study [Mean (SD)].

Variable	N(%) (n = 416)	PF	RP	BP	GH	VT	SF	RE	MH
Gender									
Male	159 (38.22%)	93.52 (12.15)	69.30 (39.62)	73.48 (20.52)	61.86 (20.63)	61.10 (17.59)	77.43 (19.58)	46.96 (43.45)	61.43 (17.29)
Female	257 (61.78%)	93.23 (8.72)	70.43 (37.71)	71.42 (19.74)	58.70 (19.01)	61.46 (16.66)	78.99 (16.88)	41.89 (42.75)	60.33 (16.91)
Major									
Clinical medicine (eight-year programme) in the 4th year	75 (18.03%)	92.07 (11.69)	61.24 ^b^ (43.05)	70.35 ^a^ (18.61)	56.71 ^a^ (20.58)	56.93 ^a^ (16.25)	77.19 ^a^ (18.17)	40.44 (42.90)	57.87 ^a^ (18.59)
Clinical medicine (five-year programme) in the 4th year	92 (22.12%)	94.08 (7.98)	68.48 (38.31)	73.49 (19.43)	59.85 (20.62)	61.03 (15.17)	79.59 (18.07)	44.93 (44.87)	60.61 (15.39)
Preventive medicine in the 4th year	85 (20.43%)	91.76 (14.88)	72.35 (38.77)	77.11 (20.00)	62.11 (18.07)	63.94 (17.85)	78.82 (15.68)	47.06 (42.82)	63.53 (15.61)
Oral medicine in the 4th year	81 (19.47%)	94.32 (7.15)	78.40 (35.30)	71.72 (20.74)	66.47 (16.65)	65.93 (15.43)	82.03 (15.87)	50.62 (43.82)	65.73 (14.35)
Clinical medicine (five-year programme) in the 3rd year	83 (19.95%)	94.34 (6.71)	68.98 (35.48)	67.92 (20.56)	54.22 (20.30)	58.43 (18.87)	74.16 (20.98)	35.74 (39.91)	55.81 (19.49)
Total	416 (100%)	93.34 (10.16)	70.00 (38.41)	72.20 (20.04)	59.91 (19.68)	61.32 (17.00)	78.39 (17.96)	43.83 (43.04)	60.75 (17.04)

^a^ One-way ANOVA, *p* < 0.1; ^b^ Kruskal-Wallis test, *p* < 0.1.

**Table 3 ijerph-16-05165-t003:** The SF-36 dimension scores of different Chinese populations [Mean (SD)].

Dimension	Medical Students (N = 416)	American Chinese (N = 156)	Taiwanese (N = 1191)	Hong Kong (N = 2410)	Sichuan (N = 2249)	Hangzhou (N = 1688)	Shanghai (N = 919)
PF	93.34 (10.16)	79.4 (23.4)	92.6 (11.5)	91.8 (12.9)	90.6 (15.4)	82.2 (19.8)	89.7 (14.8)
RP	70.00 (38.41)	67.5 (37.3)	83.6 (28.9)	82.4 (31.0)	79.5 (34.7)	81.2 (33.6)	93.8 (22.6)
BP	72.20 (20.04)	62.3 (21.9)	82.4 (16.8)	84.0 (21.9)	85.6 (18.4)	81.5 (20.5)	94.6 (13.8)
GH	59.91 (19.68)	58.8 (22.7)	67.5 (18.2)	56.0 (20.2)	69.6 (21.3)	56.7 (20.2)	68.8 (19.4)
VT	61.32 (17.00)	59.0 (20.3)	65.3 (15.2)	60.3 (18.7)	70.3 (17.1)	52.0 (20.9)	71.8 (18.3)
SF	78.39 (17.96)	75.1 (22.7)	79.4 (16.0)	91.2 (16.5)	86.9 (17.3)	83.0 (17.8)	94.3 (12.1)
RE	43.83 (43.04)	61.2 (43.7)	71.3 (37.0)	71.7 (38.4)	76.5 (38.5)	84.4 (32.4)	95.1 (20.6)
MH	60.75 (17.04)	63.9 (20.4)	68.4 (14.7)	72.8 (16.6)	72.7 (16.8)	59.7 (22.7)	81.8 (14.7)

**Table 4 ijerph-16-05165-t004:** Scores of medical students according to different impact factors of QoL in the study [Mean (SD)].

Variables	N (%) (n_2_ = 416)	PF	RP	BP	GH	VT	SF	RE	MH
Physical exercise times per week									
Never	185 (44.47%)	92.27 ^a^ (10.17)	68.48 (38.77)	69.94 (20.07)	55.50 ^a^ (19.22)	58.89 ^a^ (17.17)	77.12 (18.32)	41.62 (42.60)	58.03 ^a^ (17.24)
1–2	152 (36.54%)	93.62 (9.28)	69.90 (38.85)	73.21 (20.22)	61.39 (19.43)	63.59 (16.43)	79.17 (17.64)	45.83 (43.20)	62.18 (16.98)
3–4	63 (15.14%)	96.35 (10.21)	73.02 (37.65)	74.32 (18.91)	68.32 (18.73)	63.10 (17.40)	80.07 (17.87)	48.15 (44.31)	65.14 (16.02)
>4	16 (3.85%)	91.25 (15.11)	76.56 (34.72)	80.50 (20.32)	63.69 (19.00)	60.94 (16.45)	79.17 (17.63)	33.33 (42.16)	61.25 (15.37)
Post-exercise status									
Pleasant	158 (37.98%)	94.81 ^b^ (11.42)	78.64 ^b^ (33.01)	75.20 ^a^ (19.09)	67.54 ^a^ (18.43)	67.78 ^a^ (14.02)	80.45 (16.66)	47.68 (44.41)	66.20 ^a^ (15.34)
Somewhat invigorated	134 (32.21%)	93.54 (7.22)	68.66 (38.74)	69.99 (18.97)	56.19 (18.55)	57.80 (17.76)	79.10 (16.24)	41.79 (41.02)	58.51 (16.32)
No change	39 (9.38%)	92.05 (7.76)	66.03 (42.72)	74.49 (21.98)	56.33 (16.50)	59.10 (17.05)	78.35 (18.19)	47.86 (45.75)	59.59 (20.06)
A little exhausted	68 (16.35%)	91.18 (11.75)	59.46 (43.29)	68.43 (21.34)	53.01 (20.95)	56.25 (17.73)	73.86 (20.48)	40.20 (42.93)	53.71 (17.77)
Exhausted	17 (4.09%)	89.71 (13.40)	51.47 (35.87)	71.71 (24.39)	54.00 (20.21)	54.41 (15.80)	71.90 (26.97)	29.41 (38.88)	58.59 (13.34)
Satisfaction with your family									
Very satisfied	141 (33.89%)	93.55 ^a^ (11.33)	70.57 (38.42)	74.62 ^a^ (19.56)	64.22 ^a^ (18.54)	65.71 ^a^ (16.24)	80.69 ^b^ (17.65)	46.57 ^b^ (42.52)	64.28 ^a^ (18.18)
Relatively satisfied	198 (47.60%)	93.33 (10.08)	72.31 (37.61)	73.51 (19.93)	59.46 (19.17)	61.72 (15.75)	79.63 (15.38)	47.31 (43.16)	61.62 (15.02)
General	57 (13.70%)	94.04 (6.37)	63.16 (38.41)	64.35 (19.83)	54.39 (21.27)	51.84 (18.44)	71.15 (20.34)	30.99 (42.66)	51.86 (17.71)
Relatively dissatisfied	16 (3.85%)	93.44 (6.76)	68.75 (45.19)	64.75 (17.10)	55.19 (18.75)	55.00 (13.90)	74.31 (19.76)	33.33 (40.37)	53.25 (15.91)
Very dissatisfied	4 (0.96%)	76.25 (14.93)	37.50 (43.30)	64.00 (29.74)	27.50 (14.43)	47.50 (31.23)	55.56 (52.12)	0.00 (0.00)	50.00 (15.14)
Distressing family events in one year									
Yes	60 (14.42%)	92.33 (14.66)	61.25^c^ (42.29)	69.68 (22.45)	59.20 (20.95)	60.75 (18.29)	75.37 (22.32)	40.00 (44.60)	59.93 (17.39)
No	356 (85.58%)	93.51 (9.20)	71.47 (37.58)	72.63 (19.61)	60.03 (19.49)	61.42 (16.80)	78.90 (17.10)	44.48 (42.80)	60.89 (17.00)
Time for extracurricular research (T1)									
None	237 (56.97%)	93.12 (10.97)	69.28 (38.26)	71.47 (20.38)	59.12 (20.72)	60.49 (18.48)	79.13 (18.43)	43.32 (42.62)	60.86 (17.81)
T1 < 2 h	106 (25.48%)	93.77 (8.04)	68.40 (41.16)	73.51 (19.95)	60.04 (19.25)	62.59 (14.63)	78.41 (16.72)	41.82 (42.68)	61.81 (16.15)
2 ≤ T1 < 4 h	32 (7.69%)	93.75 (7.30)	72.66 (37.22)	73.31 (17.14)	63.09 (18.31)	65.47 (12.85)	76.39 (15.39)	47.92 (46.33)	61.75 (13.48)
4 ≤ T1 < 6 h	12 (2.88%)	91.25 (21.12)	66.67 (40.36)	71.17 (19.80)	57.00 (16.51)	60.83 (15.79)	77.78 (17.08)	55.56 (43.42)	55.00 (12.20)
T1 ≥ 6 h	29 (6.97%)	93.97 (5.88)	80.17 (29.41)	72.66 (21.62)	63.59 (14.92)	59.14 (16.91)	74.71 (21.60)	45.98 (45.79)	57.24 (18.98)
Relationship with roommates									
Very good	101 (24.28%)	93.81 (12.00)	77.72 ^b^ (34.80)	77.07 ^a^ (17.64)	61.61 ^a^ (19.62)	66.04 ^a^ (15.63)	83.94 ^b^ (15.19)	53.47 ^a^ (43.23)	67.45 ^b^ (13.42)
Good	232 (55.77%)	93.68 (9.37)	71.44 (37.45)	70.70 (19.97)	60.80 (18.91)	61.29 (16.33)	78.54 (17.56)	41.95 (42.78)	61.14 (16.43)
General	72 (17.31%)	91.74 (9.90)	56.50 (42.45)	70.81 (21.29)	56.25 (21.86)	57.36 (18.15)	73.46 (17.65)	39.81 (42.84)	52.61 (19.26)
Bad or very bad	11 (2.64%)	92.27 (9.58)	56.82 (41.97)	68.45 (28.36)	49.45 (17.62)	44.55 (20.18)	56.57 (26.97)	21.21 (34.23)	44.36 (12.58)
Sleep quality									
Very good	74 (17.79%)	95.34 ^a^ (7.82)	80.74 ^b^ (30.84)	78.00 ^a^ (16.26)	68.45 ^b^ (19.54)	69.59 ^a^ (14.16)	81.38 ^b^ (19.30)	56.31 ^b^ (44.46)	67.89 ^a^ (13.75)
Good	147 (35.34%)	94.29 (852)	78.23 (33.01)	74.04 (19.49)	63.80 (16.47)	64.63 (14.83)	81.56 (16.53)	48.30 (43.65)	64.14 (15.13)
General	126 (30.29%)	92.42 (12.72)	62.30 (41.44)	70.87 (21.35)	56.50 (20.53)	59.68 (16.39)	78.48 (14.99)	42.06 (42.20)	59.21 (17.01)
Bad or very bad	69 (16.59%)	90.87 (9.81)	54.97 (43.29)	64.51 (20.16)	48.68 (18.31)	48.41 (17.61)	68.28 (20.81)	24.15 (34.72)	48.70 (17.73)

^a^ One-way ANOVA, *p* < 0.1; ^b^ Kruskal-Wallis test, *p* < 0.1; ^c^
*t*-test, *p* < 0.1.

**Table 5 ijerph-16-05165-t005:** Standardised regression coefficients (*p* < 0.05) of the impact factors on quality of life resulting from multivariate stepwise regression.

Variable	PF	RP	BP	GH	VT	SF	RE	MH
Clinical medicine (eight-year programme) in the 4th year	NI	NS	NS	NS	NS	NS	NI	NS
Clinical medicine (five-year programme) in the 4th year	NI	NS	NS	NS	NS	NS	NI	NS
Preventive medicine in the 4th year	NI	NS	0.124	NS	0.113	NS	NI	0.131
Oral medicine in the 4th year	NI	NS	NS	0.136	0.124	NS	NI	0.144
Clinical medicine (five-year programme) in the 3rd year	NI	NS	NS	NS	NS	NS	NI	NS
Physical exercise times per week	NI	NI	NI	0.160	NS	NI	NI	NS
Post-exercise status	0.122	0.145	NS	0.136	0.150	NI	NI	0.126
Satisfaction with your family	NS	NI	0.103	0.106	0.130	0.107	NS	0.101
Relationship with roommates	NI	0.113	NS	NS	0.124	0.209	NS	0.240
Sleep quality	0.121	0.189	0.175	0.250	0.269	0.138	0.223	0.222

NI, Not Involved; NS, Not Significant (*p* > 0.05).Variable coding: Major [Clinical medicine (five-year programme) in the 4th year = 1, the others = 0; Preventive medicine in the 4th year = 1, the others = 0; Oral medicine in the 4th year = 1, the others = 0; Clinical medicine (five-year programme) in the 3rd year = 1, the others = 0]; Physical exercise times per week (never = 1; 1–2 times = 2; 3–4 times = 3; >4 = 4); Post-exercise status (pleasant = 5; somewhat invigorated = 4; no change = 3; a little exhausted = 2; exhausted = 1), Satisfaction with your family (very satisfied = 5; relatively satisfied = 4; general = 3; relatively dissatisfied = 2; very dissatisfied = 1); Relationship with roommates (very good = 4; good = 3; general = 2; bad or very bad = 1); Sleep quality (very good = 4; good = 3; general = 2; bad or very bad = 1).

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
