# Peer review of "Health-Related Quality of Life of Medical Students in a Chinese University: A Cross-Sectional Study"

_ijerph, 2019, doi:10.3390/ijerph16245165_

Round 1
Reviewer 1 Report
This paper conducted a randomized controlled trial to explore the feasibility of the mandarin SF-36 in assessing the health-related quality of life (HRQOL) of medical students and to assess HRQOL of medical students at SunYat-sen University while aiming to find the associated factors that caused the variation among these students. They suggested that the mandarin SF-36 was reliable for measuring the HRQOL and that the HRQOL of medical students in university China was relatively poor and its improvement requires concerted efforts. I do have some comments as listed below in the order noted.
Comment 1:
The quality of the data set is very important, especially in a randomized controlled trial study. For this reason, please clarify the included criteria and excluded criteria of sample collection in the Materials and Methods section and please also provide a flowchart immediately at the subsection of data collection.
Comment2:
Please provide the norms of each dimension of the SF-36 in China and compare the differences of each dimension of the SF-36 between the norms and the medical students.
Comment 3:
Please also provide the P values from multivariate stepwise regression in Table 4.
Author Response
Comment 1: The quality of the data set is very important, especially in a randomized controlled trial study. For this reason, please clarify the included criteria and excluded criteria of sample collection in the Materials and Methods section and please also provide a flowchart immediately at the subsection of data collection.
Response: We thank the reviewer for the valuable suggestions. The included criteria and excluded criteria of sample collection are also important for a cross sectional study. And we have mentioned the included criteria of the sample collection in the Materials and Methods section. Students were included from the following five different majors and years: eight-year medical program students in the 4th year, preventive medicine in the 4th year, oral medicine in the 4th year, five-year medical program students in the 3rd and the 4th year.
For the excluded criteria, we have agreed that we omitted it. In fact, we excluded students who dropped out of school or changed major. And we have added it to the Materials and Methods section. We have also added a flow chart of data collection, organization and analysis of data (Figure 1) in the study.
Comment 2: Please provide the norms of each dimension of the SF-36 in China and compare the differences of each dimension of the SF-36 between the norms and the medical students.
Response: We thank the reviewers for the constructive comment. We have added the comparison about the level of QoL between medical students in the study and other Chinese populations including Shanghai, Hangzhou, Sichuan, Hong Kong, Taiwan and American Chinese populations. And corresponding discussion have been showed in Discussion section.
We have written the following in the manuscript.
“Table 3 and Figure 2 compare the eight SF-36 dimension scores among different Chinese populations, including American Chinese, Taiwanese, Hong Kong, Sichuan, Hangzhou, and Shanghai. Medical students involved in the survey scored highest in the PF dimension and lowest in the RE dimensions. The MH dimension score of medical students was close to that of the Hangzhou population, ranking the lowest. The SF, BP, and RP dimension scores of Taiwanese, Hong Kong, Sichuan, Hangzhou, and Shanghai populations were higher than those of medical students involved in the survey, who scored lower than Taiwanese, Sichuan, and Shanghai populations in the VT and GH dimensions.” (page 4, lines 171-177)
“Except for the highest PF dimension score (93.34), the medical students had the lowest scores in the RE and MH dimensions, relatively lower scores in the SF/BP/RP dimensions, and some moderate scores in the VT/GH dimensions.” (page 6, lines 230-232)
Comment 3: Please also provide the P values from multivariate stepwise regression in Table 4.
Response: We have added the P value (P<0.05) of standardized regression coefficients in the label of Table 5 (used to be Table 4). And in this revision, we have changed the setting of a in the ANOVA, T-test and Kruskal-Wallis test, from a=0.05 to a=0.10, to avoid the missing of potential important impact factors. Intriguingly, this change didn’t change the final result of multivariate stepwise regression. We have also filled the left blanks in Table 5 with NI (Not Involved) and NS (Not Significant) to avoid confusion.
Reviewer 2 Report
Dear Authors,
This is a well-written paper. I just have few suggestions and comments as follow:
Please correct these typos: Do not use contraction for auxiliary verbs. For example: page 1 line 44: "it's" should be written as it is; page 4 line 138 "don't " should be written as "do not" etc. Please check and make necessary amendment throughout the manuscript. page 3 line 111,112,130,131 and heading of Table 1; page 8 line 185,186: Cronbach's A or Cronbach's alpha NOT Chronbach's a page 3 line 132 and heading of Table 1; page 8 line 188: Sperman's correlation NOT Spearman correlation page 4: change all "means" to "mean Please check numbering page 2 line 66 should be numbered 2 NOT 1 page 2 line 84 should be numbered 2.2 NOT 1.2 page 3 line 101 should be numbered 2.3 NOT 1.2 page 3 line 125 should be numbered 3 NOT 1 page 8 line 176 should be numbered 4 NOT 2 Please correct the formatting for table 1,2,3 and 4. Please make sure the headings and the tables are in one page (not separated in 2 different pages) and all figures in the tables are align. Results: Authors can use bar chart to illustrate the comparison between 5-year and 8-year programs for each domain of SF-36 in Table 2 variable coding for Table 4 is quite confusing. Is there anything you could do to improve readability and reader's understanding? Discussion More critical discussion and explanation needed when comparing the results with previous studies. For example, participants in this study had lower mean for certain dimensions compared with the previous study (Wang et al., 2008). Why? justification error for the paragraph "Strength and limitations of this study" on page 10, line 262-268Author Response
Comment 1: Please correct these typos: Do not use contraction for auxiliary verbs. For example: page 1 line 44: "it's" should be written as it is; page 4 line 138 "don't " should be written as "do not" etc. Please check and make necessary amendment throughout the manuscript. page 3 line 111,112,130,131 and heading of Table 1; page 8 line 185,186: Cronbach's A or Cronbach's alpha NOT Cronbach's a page 3 line 132 and heading of Table 1; page 8 line 188: Spearman's correlation NOT Spearman correlation page 4: change all "means" to "mean Please check numbering page 2 line 66 should be numbered 2 NOT 1 page 2 line 84 should be numbered 2.2 NOT 1.2 page 3 line 101 should be numbered 2.3 NOT 1.2 page 3 line 125 should be numbered 3 NOT 1 page 8 line 176 should be numbered 4 NOT 2
Response: We thank the reviewer for the detailed suggestion. All have been done now. And “Cronbach’s a” and “Spearman correlation” have been replaced by “Cronbach’s α” and “Spearman’s correlation” respectively.
Comment 2: Please correct the formatting for table 1,2,3 and 4. Please make sure the headings and the tables are in one page (not separated in 2 different pages) and all figures in the tables are align. Authors can use bar chart to illustrate the comparison between 5-year and 8-year programs for each domain of SF-36 in Table 2. Variable coding for Table 4 is quite confusing. Is there anything you could do to improve readability and reader's understanding?
Response: We thank the reviewer for the suggestions. We have adjusted the formatting for these tables. And we have used “[Insert xxx about here]” to show the tables and figures positions and made the detailed tables and figures in the end of the manuscript with a heading along with the word “continued” in the second page, when the table was separated in 2 different pages.
The suggestion for bar chart is enlightening. Although we don’t use it to illustrate the comparison between 5-year and 8-year programs for each domain of SF-36 in Table 2, we have added a bar chart to compare the QoL level of medical students in the study and six other Chinese populations to improve reader’s understanding.
We have adjusted the variable coding for Table 4 and filled the left blanks with “NI” and “NS” to avoid confusion---see below. We have also retained a native English speaker in the field to proofread the revised manuscript to improve readability and clarity.
“NI: Not Involved; NS: No Significant (P>0.05).
Variable coding: Major [Clinical medicine (five-year programme) in the 4th year =1, the others=0; Preventive medicine in the 4th year =1, the others=0;Oral medicine in the 4th year =1, the others=0; Clinical medicine (five-year programme) in the 3rd year =1, the others=0], Physical exercise times per week (never=1; 1~2 times=2; 3~4 times=3; >4=4), Post-exercise status (pleasant=5; somewhat invigorated=4; no change=3; a little exhausted=2; exhausted=1), Satisfaction with your family (very satisfied=5; relatively satisfied=4; general=3; relatively dissatisfied=2; very dissatisfied=1), Relationship with roommates (very good=4; good=3; general=2; bad or very bad=1), Sleep quality (very good=4; good=3; general=2; bad or very bad=1).”
Comment 3: More critical discussion and explanation needed when comparing the results with previous studies. For example, participants in this study had lower mean for certain dimensions compared with the previous study (Wang et al., 2008). Why?
Response: We thank the reviewer for the valuable suggestion. In the revised manuscript, we have added the data for the comparison between medical students in the study and six other Chinese populations in the Result section and further discussed the results of the comparison in the Discussion section.
We have written the following in the Discussion section.
“Intriguingly, though the medical students had the highest score in the PF dimension. This may be influenced by the research population in the survey, consisting of college students, who tend to have higher physical function than other populations, including the elderly. The RE dimension, which is often considered one of the best measures of MCS, scored the lowest among the eight dimensions in our study, at 43.83 (43.04). The value was lower than that of other Chinese populations, and this fact demonstrated that the quality of life of medical students in our study was fairly poor. This may be attributed to high workloads, heavy academic pressure, a lower household income, and low sleep quality. Zhong et al. reported that one-third of medical students undergoing postgraduate neurology specialty training in China showed symptoms of depression, and those without such symptoms had significantly higher QoL scores. Accurate measures should be taken by medical schools to ensure emotional support for their students, thus improving the QoL scores.” (page 6, lines232-242)
Comment 4: justification error for the paragraph "Strength and limitations of this study" on page 10, line 262-268
Response: We thank the reviewer for the useful suggestion. We have adjusted the “Strength and limitations of this study” section---see below.
“Certain flaws remained. For instance, detailed information on non-responders was not collected, nor were we sure whether there were differences between responders and non-responders. Further, our sample had an adequate size but was recruited from only one university due to limited funds, which also limited the generalisation of our results, and further studies need to be done. Moreover, due to the inherent limits of a cross-sectional study, timing between independent and dependent variables was not under strict control. It should also be noted that certain option-setting about some impact factors focused more on participants’ subjective feelings than on precise and objective indices, such as the level of exercise and sleep quality. This kind of option-setting benefited the participants to better complete the questionnaires in a limited time. Further studies on the quality of life are still required, where more precise and objective indices can be applied.” (page 7-8,lines 310-318)
Reviewer 3 Report
Overall comments: The idea of measuring health-related quality of life among medical students is very reasonable. It is a timely topic as the mental health and burnout of professional students is of particular interest at this time.
In general, the reliability of the SF-36 survey was mentioned several times though-out the article. This is questionable as there was no evidence presented to ensure reliability of the survey in predicting HRQOL for medical students. Additionally, would have appreciated the addition of a comparison group to determine the differences in scores for a different population of students.
There were also numerous grammatical errors throughout the article which could have contributed to some of the confusion while reading the study. Would recommend an editor of the English language to very thoroughly review the translation to ensure appropriate flow for the readers.
There is no mention of review by an ethics board or informed consent from the participants.
I have the following specific recommendations:
Introduction section: An explanation of the SF-36 survey, including background information and explanation of the different categories and interpretation of the scores would help the reader better understand the results of this study.
Materials and methods: Line 71: sample size was arbitrarily and conveniently defined- these words are misleading likely due to an inappropriate translation. Consider changing to “convenience sample” and removing the word “arbitrarily.” It is also unclear what is meant by “three classes from each major” were these students in different years? If not, it would appear that the same group of students was sampled 3 times.
The specific questions asked of the students (exercise, post-exercise status, family satisfaction, and so on) and their planned analysis are not mentioned in the methods.
Results: Line 164: better post-exercise status- you define this status later in the article but consider defining when you first mention the phrase. Unclear what this status measures. Table at bottom of page 7: unclear whether there may or may not be some data points missing. If intentionally left blank, consider providing explanation of comparator groups/explain why no data listed here.
Author Response
Comment 1: the reliability of the SF-36 survey was mentioned several times though-out the article. This is questionable as there was no evidence presented to ensure reliability of the survey in predicting HRQOL for medical students.
Response: We thank the reviewer for the comment. We have used Cronbach’s α coefficients to present the reliability of the SF-36 survey. Cronbach’ α coefficients of seven dimensions including PF, RP, BP, GH, VT, RE and MH were over 0.7. The SF dimension had a low Cronbach’sαcoefficients with a value of 0.481. Generally, the result showed a good internal consistency of the SF-36 for the purpose in the study. The low Cronbach's alpha level of the SF scale was also found in previous studies in China and we have explained it in Discussion section. And we have also added the split-half reliability to support the reliability of the SF-36 survey.
We have clarified this in the Introduction, Result and Discussion section.
“The Mandarin SF-36 has been widely used in the QOL measurement of general Chinese populations, and the reliability and validity of this form in Chinese medical students have been verified. Therefore, the need to assess QOL of medical students by means of the SF-36 is justified” (page2, lines 63-66)
“Split-half reliability was calculated by comparing the scores of the odd half with those of the even half in each SF-36 dimension. Cronbach’s α coefficient assessed the internal consistency of the SF-36 questionnaire, and a Cronbach’s α coefficient not less than 0.7 was generally considered sufficient to demonstrate internal consistency.” (page 3, lines 131-134)
“In seven of eight dimensions, the spilt-half reliability coefficients valued more than 0.8 and Cronbach’s α coefficients valued more than 0.7. This showed a good internal consistency of the SF-36 for the purposes of the study. But the SF dimension had a low split-half reliability coefficient and Cronbach’s α coefficient of less than 0.7. Previous studies have reported similar issues. It may be the result of an unclear conceptualization of social function in the Mandarin SF-36 and certain misunderstandings caused by differences in cultures. In China, "social activities", refer not only to everyday life with someone with whom they are familiar in informal situations, but also to formal activities with other people, such as joining a new department or attending a conference. Individuals occasionally find themselves required to participate in formal activities in the face of a certain degree of ill health or a bad mood. Misunderstandings may result in low Cronbach’s α levels.” (page 5 lines 217-225)
Comment 2: Additionally, would have appreciated the addition of a comparison group to determine the differences in scores for a different population of students.
Response: We thank the reviewer for the valuable comment. We have added the comparison between medical students in the study and six other Chinese populations in the Result section and further discussed the results of the comparison in the Discussion section.
We have added the following in the revised manuscript.
“Table 3 and Figure 2 compare the eight SF-36 dimension scores among different Chinese populations, including American Chinese, Taiwanese, Hong Kong, Sichuan, Hangzhou, and Shanghai. Medical students involved in the survey scored highest in the PF dimension and lowest in the RE dimensions. The MH dimension score of medical students was close to that of the Hangzhou population, ranking the lowest. The SF, BP, and RP dimension scores of Taiwanese, Hong Kong, Sichuan, Hangzhou, and Shanghai populations were higher than those of medical students involved in the survey, who scored lower than Taiwanese, Sichuan, and Shanghai populations in the VT and GH dimensions.” (page 4, lines 171-177)
“Our study found that the quality of life of medical students was relatively poor when compared with each dimension of the SF-36 in different Chinese populations. Except for the highest PF dimension score (93.34), the medical students had the lowest scores in the RE and MH dimensions, relatively lower scores in the SF/BP/RP dimensions, and some moderate scores in the VT/GH dimensions. Intriguingly, though the medical students had the highest score in the PF dimension. This may be influenced by the research population in the survey, consisting of college students, who tend to have higher physical function than other populations, including the elderly. The RE dimension, which is often considered one of the best measures of MCS, scored the lowest among the eight dimensions in our study, at 43.83 (43.04). The value was lower than that of other Chinese populations, and this fact demonstrated that the quality of life of medical students in our study was fairly poor. This may be attributed to high workloads, heavy academic pressure, a lower household income, and low sleep quality. Zhong et al. reported that one-third of medical students undergoing postgraduate neurology specialty training in China showed symptoms of depression, and those without such symptoms had significantly higher QoL scores [22]. Accurate measures should be taken by medical schools to ensure emotional support for their students, thus improving the QoL scores.” (page 6, lines 229-242)
Comment 3: There were also numerous grammatical errors throughout the article which could have contributed to some of the confusion while reading the study. Would recommend an editor of the English language to very thoroughly review the translation to ensure appropriate flow for the readers
Response: We thank the reviewer for the useful suggestion. We have retained the services of a native English speaker in the field to proofread the revised manuscript to improve readability and clarity.
Comment 4: There is no mention of review by an ethics board or informed consent from the participants.
Response: We thank the reviewer for the important reminding. We have supplemented them in the Materials and Methods section---see below.
“All participants gave their informed consent for inclusion before being accepted into the study. The study was conducted in accordance with the Declaration of Helsinki, and the protocol was approved by the Ethics Committee of the school of public health at Sun Yat-sen University.” (page 3, lines 115-117)
Comment 5: Introduction section: An explanation of the SF-36 survey, including background information and explanation of the different categories and interpretation of the scores would help the reader better understand the results of this study.
Response: We thank the reviewer for this valuable suggestion. We have showed the background of the SF-36 survey in the Introduction section and added something more to help them to understand more about this survey. Considering the length of the Introduction section, we have made the interpretation of the categories and scores of SF-36 at the part of “Materials and Methods”.
The details have been showed below.
“In 1991, the international quality-of-life assessment formulated the standard procedures of the SF-36 to unify its use in various countries. The SF-36 questionnaire was derived from the Medical Outcomes Study (MOS) in the Boston Health Research Institute of the United States in 1989, an instrument with 149 items. The initial objective of the MOS was to evaluate the medical decisions and patient outcomes under the influence of different systematic health care approaches. However, its simplified version was extended to assess the health-related quality of life and finally became the SF-36 questionnaire used today. Since then, the SF-36 questionnaire has been a widely validated and popular tool used in the assessment of quality of life among the general population ages 14 years and older.” (page 2, lines 50-58)
“The Chinese version of the SF-36 measures eight health-related domains, including physical function (PF, limitations in physical activities because of health problems), physical role (RP, limitations in usual role activities because of physical health problems), body pain (BP), general health (GH, general health perceptions), vitality (VT, energy and fatigue), social function (SF, limitations in social activities because of physical or emotional problems), emotional role (RE, limitations in usual role activities because of emotional problems), mental health (MH, psychological distress and well-being), and one single-item dimension on health transition. The physical component summary (PCS) consists of PF, RP, BP, and GH, while the mental component summary (MCS) consists of VT, SF, RE, and MH.” (page 3, lines 97-105)
“Higher scores suggested a higher level for the QoL.” (page 3, line129)
Comment 6: Materials and methods: Line 71: sample size was arbitrarily and conveniently defined- these words are misleading likely due to an inappropriate translation. Consider changing to “convenience sample” and removing the word “arbitrarily.” It is also unclear what is meant by “three classes from each major” were these students in different years? If not, it would appear that the same group of students was sampled 3 times.
Response: We thank the reviewer for the useful suggestion. All have been done now.
Comment 7: The specific questions asked of the students (exercise, post-exercise status, family satisfaction, and so on) and their planned analysis are not mentioned in the methods.
Response:We thank the reviewer for the comment. We have mentioned these factors and their planned analysis in the Materials and Methods section. Although we don’t showed the specific questions about these factors, we have added our questionnaire containing the specific questions about these factors in supplement files.
We have written the followings in the Materials and Methods section.
“An f test was also calculated to evaluate the homogeneity of variance. Based on the homogeneity of variance, one-way analysis of variance and a t test were applied to compare the means of the SF-36 and its component scores according to different impact factors. Otherwise, the Kruskal-Wallis test would be applied. We set α = 0.1 to avoid missing potential important factors. Impact factors differing among at least one of the eight dimensions were selected to make an analysis of multivariate stepwise regression in the corresponding dimensions.” (page 3-4, lines 137-141)
Comment 8: Results: Line 164: better post-exercise status- you define this status later in the article but consider defining when you first mention the phrase. Unclear what this status measures.
Response: We thank the reviewer for the useful comment. We agree that it really makes the status unclear if don’t define the status in the place where it first was mentioned. In the revision, we have defined post-exercise status to where this concept first appeared accordingly. Post-exercise status could help us know the subjective feeling of participants after exercise. The modified version is as follows:
“Post-exercise status, indicating participants’ subjective feelings of both physical and mental status after exercise, had a significant influence on the dimensions of PF, RP, BP, GH, VT, and MH.” (page 4,lines 182-184)
Comment 9: Table at bottom of page 7: unclear whether there may or may not be some data points missing. If intentionally left blank, consider providing explanation of comparator groups/explain why no data listed here.
Response: We thank the reviewer for the valuable suggestion. We have filled the left blanks in Table 5 with NI (Not Involved) and NS (Not Significant) to avoid confusion. And in this revision, we have changed the setting of a in the ANOVA, T-test and Kruskal-Wallis test, from a=0.05 to a=0.10, to avoid the missing of potential important impact factors. Intriguingly, this change didn’t change the final result of multivariate stepwise regression.
Round 2
Reviewer 3 Report
Thank you for the revisions.